# A Metabolic Change towards Fermentation Drives Cancer Cachexia in Myotubes

**DOI:** 10.3390/biomedicines9060698

**Published:** 2021-06-20

**Authors:** Michele Mannelli, Tania Gamberi, Francesca Magherini, Tania Fiaschi

**Affiliations:** Dipartimento di Scienze Biomediche, Sperimentali e Cliniche “M. Serio”, Università degli Studi di Firenze, Viale Morgagni 50, 50134 Firenze, Italy; michele.mannelli@student.unisi.it (M.M.); tania.gamberi@unifi.it (T.G.); francesca.magherini@unifi.it (F.M.)

**Keywords:** cancer, cachexia, metabolism, lactate dehydrogenase

## Abstract

Cachexia is a disorder associated with several pathologies, including cancer. In this paper, we describe how cachexia is induced in myotubes by a metabolic shift towards fermentation, and the block of this metabolic modification prevents the onset of the cachectic phenotype. Cachectic myotubes, obtained by the treatment with conditioned medium from murine colon carcinoma cells CT26, show increased glucose uptake, decreased oxygen consumption, altered mitochondria, and increased lactate production. Interestingly, the block of glycolysis by 2-deoxy-glucose or lactate dehydrogenase inhibition by oxamate prevents the induction of cachexia, thus suggesting that this metabolic change is greatly involved in cachexia activation. The treatment with 2-deoxy-glucose or oxamate induces positive effects also in mitochondria, where mitochondrial membrane potential and pyruvate dehydrogenase activity became similar to control myotubes. Moreover, in myotubes treated with interleukin-6, cachectic phenotype is associated with a fermentative metabolism, and the inhibition of lactate dehydrogenase by oxamate prevents cachectic features. The same results have been achieved by treating myotubes with conditioned media from human colon HCT116 and human pancreatic MIAPaCa-2 cancer cell lines, thus showing that what has been observed with murine-conditioned media is a wide phenomenon. These findings demonstrate that cachexia induction in myotubes is linked with a metabolic shift towards fermentation, and inhibition of lactate formation impedes cachexia and highlights lactate dehydrogenase as a possible new tool for counteracting the onset of this pathology.

## 1. Introduction

Cachexia is a complex multi-organ syndrome characterized by progressive body weight loss associated with several chronic or end-stage diseases [1,2]. Particularly, cancer cachexia affects 50–80% of patients and accounts for about 20% of cancer deaths [3]. Cachectic patients show a drastic worsening of both prognosis and quality of life, as well as a reduced tolerance and response to antineoplastic treatments [4]. In skeletal muscle, cachexia induces wasting and atrophy due to several metabolic alterations [3,5,6]. Muscle loss is due to the great increase in protein degradation, not counterbalanced by adequate protein synthesis, in which activation of the ubiquitin-dependent proteasome pathway plays a crucial role [7,8]. Many intracellular signals are involved in protein turnover leading to muscle wasting [9], such as those signaling pathways activated by inflammatory cytokines, such as tumor necrosis factor alpha [10], interleukin-1 (IL-1) [11], and IL-6 [12], secreted by either immune or tumor cells. Among cytokines, IL-6 exerts a prominent role as cachexia inductor. Indeed, IL-6 secreted by tumor cells promotes the activation of the JAK–STAT signaling pathway, leading to suppression of protein synthesis and muscle wasting [12]. In agreement, inhibition of the JAK/STAT3 pathway impedes muscle wasting in the mouse model of colon carcinoma [13].

Cancer cachexia is considered an energy-balance disorder caused by to pronounced metabolic alterations. Cancer patients have resting energy expenditure higher than healthy individuals [14], probably induced by inflammatory stimuli [15]. Furthermore, cachectic muscles show increased expression of uncoupling proteins that, dissipating the proton gradient along the inner mitochondrial membrane, leads to decreased ATP generation [3,16,17].

Despite the evidence suggesting that cancer-induced cachexia is associated with broad metabolic alterations, the potential role of muscle metabolic abnormalities in the activation and development of cancer cachexia has been little studied so far. In this regard, we report that a metabolic change towards fermentation leads myotubes towards cachexia and the inhibition of this metabolic shift impedes the cachectic process. For this aim, we used conditioned medium (CM) obtained culturing murine carcinoma cell lines (breast 4T1 and colon CT26, respectively) that is reported to be able to induce (CT 26) or not induce (4T1) cachexia in mouse model [18]. We observed that myotubes treated with CM CT26 show a cachectic phenotype associated with a decreased oxidative phosphorylation and increased lactate production. Inhibition of glycolysis using 2-deoxy-D-glucose (2-DG) or lactate dehydrogenase (LDH) by oxamate reverts the metabolic change and blocks cachexia induction. These findings suggest that inhibition of lactate production by myotubes could be a new therapeutic target for blocking activation of cancer cachexia, at least in the early phase of pathology.

## 2. Experimental Section

### 2.1. Materials

C2C12 murine myoblasts were a gift of Dr. P. Porporato, University of Turin, Italy. The 4T1, CT26, A2780, MDAMB231, A375, HCT116, and MiaPaCa-2 cell lines were from ATCC. Unless differently specified, all reagents were obtained from Sigma-Aldrich, Inc. (St. Louis, MO, USA); SDS-PAGE materials and ECL detection reagents were from Bio-Rad Laboratories, (Hercules, USA); anti-ubiquitin (sc-8017) and anti-Pyruvate Dehydrogenase (PDH)-E1 (sc-377092) primary antibodies were from Santa Cruz; IL-6 primary antibodies (#500-p56) were from Peprotech (London, UK); IL-6 was from Biovision (Milpitas, CA, USA); Tetra-methyl-rhodamine methyl ester (TMRM) probes was from Molecular Probe (Eugene, OR, USA); K-LATE kit for lactate assay was from Megazyme (Bray, Ireland); [^3^H] 2-deoxy-glucose was from Perkin Elmer (Waltham, MA, USA).

### 2.2. Cell Culture

Murine C2C12 myoblasts were cultured in growing medium composed of Dulbecco’s modified Eagle’s medium (DMEM, #ECB7501 Euroclone, Milan, Italy) supplemented with 10% fetal bovine serum in 5% CO_2_ humidified atmosphere. For differentiation, sub-confluent C2C12 were shifted from growing to differentiating medium composed of DMEM containing 2% Horse Serum (HOS). All cell lines used (4T1, CT26, A2780, MDAMB231, A375, HCT116, and MiaPaCa-2) were cultured in DMEM supplemented with 10% fetal bovine serum in 5% CO_2_ humidified atmosphere.

### 2.3. Conditioned Media

Carcinoma cells were cultured in DMEM containing 10% fetal bovine serum until 80% confluence. Then, the medium was replaced with serum free DMEM for 48 h. Culture medium, which became conditioned medium (CM) from the secretome of tumor cells, was centrifuged at 13,000 rpm, for 10 min, to eliminate debris. CM was then diluted in differentiating medium at 20% final and used for myotube treatment. CMs were used at 20% final since lower CM dilution did not induce any cachectic effect, while higher CM concentration led progressively to cell death.

### 2.4. Immunoblot Analysis

Cells were lysed for 20 min on ice in 500 µL of complete radio-immunoprecipitation assay (RIPA) buffer (150 mM NaCl, 100 mM NaF, 2 mM EGTA, 50 mM Tris HCl pH 7.5, 1 mM orthovanadate, 1% triton, 0.1% SDS, and 0.1% protease inhibitor cocktail). Lysates were clarified by centrifugation, and total protein contents were obtained by using Bradford assay (Bio-Rad Laboratories, Hercules, USA). For IL-6 detection by immunoblot, 6 mL of CM from CT26 and 4T1 cell lines were concentrated by using Amicon Ultra-4 centrifugal filter units (Millipore Sigma, St. Louis, USA) until 100 µL. Then, 20 µg of total proteins for each sample were separated by SDS-PAGE and transferred onto PVDF membranes. PVDF membranes were incubated in 2% milk, probed with primary antibodies, and incubated with secondary antibodies conjugated with horseradish peroxidase. Quantification of bands was achieved by using ImageJ software.

### 2.5. Glucose Uptake

Glucose uptake was performed by using [^3^H] 2-deoxy-glucose (0.5 mCi/mL, final concentration) diluted in a buffered solution (140 mM NaCl, 20 mM HEPES/Na, 2.5 mMMgSO4, 1 mMCaCl2, and 5 mM KCl, pH 7.4) for 15 min at 37 °C. Cells were subsequently washed with cold PBS (ECB4004, Euroclone, Milan, Italy) and lysed with 0.1 M NaOH. Incorporated radioactive glucose was assayed by scintillation counter and the obtained value was then normalized on total protein content.

### 2.6. Oxygen Consumption Assay

Myotubes were treated with CMs for 24 h. Regarding 2-DG and oxamate treatments, 2-DG (1 mM final) and oxamate (75 mM final) were added to the cells together with CMs and maintained for 24 h. Myotubes were detached, washed with PBS, and suspended in 1 mL growing medium. Cell suspension was transferred to an airtight chamber maintained at 37 °C. Oxygen consumption was measured by using a Clark-type O_2_ electrode (Oxygraph Hansatech). Oxygen content was monitored for 10 min. During the time of the assay, the cells utilized intracellular respiratory substrates that were produced and accumulated during the period of the treatment. The rate of decrease in oxygen content was taken as an index of the respiratory ability. This value was then normalized on total protein content.

### 2.7. PDH Activity

PDH activity was assayed in cell lysates by using PDH Activity Assay Kit (#MAK183, Sigma Aldrich, St. Louis, USA) according to the manufacturer’s instruction. The 2-DG (1 mg/mL, final) and oxamate (75 mg/mL, final) were added to myotubes together with CMs and maintained along the experiment. PDH activity (nmol/min/mL) was normalized on total protein content in each sample and reported in the bar graph.

### 2.8. Lactate Assay

Lactate amount was assayed in the cell medium, using K-LATE kit (Megazyme, (Bray, Ireland)) according to the manufacturer’s instructions. Prior to the treatment of myotubes, lactate amount was measured in CMs. To obtain the amount of lactate exclusively produced by myotubes, lactate quantity in CMs was subtracted from lactate in the medium obtained from treated myotubes. The obtained value has been normalized on total intracellular protein content and reported in the bar graph.

### 2.9. Confocal Analysis

C2C12 myoblasts were grown until sub-confluence on glass coverslips and then differentiated for four days. Analysis of mitochondrial membrane potential was performed by treating viable myotubes with TMRM probe (1 µM final) and with DAPI (10 µM final) for nuclei labeling for 15 min at 37 °C and immediately observed. In all experiments, emitted fluorescence was analyzed by using a confocal fluorescence microscope Leica TCS SP8.

### 2.10. Statistical Analysis

Data are presented as mean ± SD from at least three independent experiments. Statistical analysis of the data was performed by Student’s *t*-test or by one-way ANOVA, using Graph Pad Prism (Graphpad Holdings, LLC, USA), version 6.0. A *p*-value < 0.05 was considered statistically significant.

## 3. Results

### 3.1. The Treatment with CM CT26 Induces a Metabolic Modification in Myotubes

We wanted to address whether induction of cancer cachexia is associated/induced to a metabolic change in myotubes. For this aim, myotubes originated by differentiation of C2C12 murine myoblasts were treated with conditioned medium (CM) obtained cultivating murine colon carcinoma CT26 and murine breast carcinoma 4T1 cell lines. We chose 4T1 and CT26 cell lines as carcinoma cell models, unable and able, respectively, to induce cachexia in the mouse model, as already reported [18].

To evaluate the induction of cancer cachexia in myotubes by CMs, we firstly examined the phenotypic effects induced by CM CT26 and CM 4T1. Results show that CM-CT26-treated myotubes display muscle wasting phenotype characterized by a significant decrease in myotube width when compared to control and CM-4T1-treated myotubes (Appendix A). The same result was obtained by confocal analysis. Indeed, CM CT26 clearly impairs myotube phenotype, since myotubes appear thinner in comparison to control and CM-4T1-treated myotubes (Appendix A). Skeletal muscle wasting is mainly driven by a dramatic promotion of catabolic pathways, leading to an unbalance between protein synthesis and degradation [19]. Particularly, the ubiquitin-dependent proteasome pathway has clearly emerged as one of the main degradative system involved in the muscle protein breakdown leading to cachexia [20]. In agreement, we report that CM-CT26-mediated phenotypic effect on myotubes is associated with the enhancement of ubiquitin-proteasome system, as highlighted by the increase in total protein ubiquitination (Appendix A). Conversely, autophagy is not activated by CM CT26 as shown by anti-LC3 immunoblot (Appendix A).

Hence, we planned to study carbohydrate metabolism in myotubes exposed to CM 4T1 and CM CT26. In this context, glucose uptake and lactate measurement represent useful tools to detect shift in glucose catabolism, while myotube oxygen consumption provides information about mitochondrial oxidative phosphorylation. Glucose uptake analysis shows that CM CT26 treatment markedly enhances myotube glucose assumption when compared to control and CM-4T1-treated myotubes (Figure 1A). To assess the effects of CMs on mitochondrial respiration, we examined myotube Oxygen Consumption Rate (OCR). The results show that CM-CT26-treated myotubes have a lower OCR with respect to control and CM-4T1-treated myotubes (Figure 1B). The assay of lactate production in myotube culture medium evidence that CM CT26 strongly increases myotube lactate production in comparison to control and CM-4T1-treated myotubes (Figure 1C).

The observation that CM CT26 treatment greatly affects oxygen consumption in myotubes suggests that CM CT26 could induce significant alterations in mitochondria. To verify this, we assay mitochondrial membrane potential by using TMRM, a cell-permeant dye that accumulates in active mitochondria with intact membrane potential and decreases upon loss of potential. Confocal images show that myotubes treated with CM CT26 have altered mitochondrial membrane potential, since TMRM fluorescence is decreased of about 35% in comparison with control and CM 4T1 treated myotubes (Appendix A). The use of the different mitochondrial probe JC1 leads to the same results. JC1 differently stains mitochondria based on their membrane potential. Healthy mitochondria are red colored, while altered mitochondria appear green stained. As already observed with TMRM probe, mitochondria in CT26-treated myotubes show altered mitochondrial membrane potential (green stained) while mitochondria of control and CM 4T1-treated myotubes appear red colored (Appendix A), as confirmed by the ratio between red and green fluorescence (Appendix A). Moreover, CM-CT26-treated myotubes show decreased expression level of the OXPHOS complexes in the inner mitochondrial membrane in comparison with control and CM-4T1-treated cells (Appendix A). Finally, immunoblot analysis of citrate synthase level, normally used as a marker of mitochondria amount [21] shows similar enzyme level in each condition examined, thus suggesting that CM CT26 treatment does not affect mitochondria quantity (Appendix A).

These findings suggest that CM CT26 mediates a metabolic shift towards fermentation in myotubes, enhancing glucose uptake and the conversion of glucose to lactate in aerobic conditions. In addition, CM CT26 induces in myotubes significant alterations in mitochondria, ranging from modification of mitochondrial membrane potential to the decreased level of OXPHOX complexes, thus suggesting that these alterations could be involved in the decreased oxygen consumption detected in CM-CT26-treated myotubes.

### 3.2. Inhibition of Glycolysis or Lactate Production Prevents the CM-CT26-Induced Cachexia in Myotubes

Although the molecular mechanisms underlying cancer cachexia are widely studied [5], the possible role of metabolic changes in the onset of cachexia is unexplored so far. Thus, we planned to elucidate the possible involvement of the metabolic shift towards fermentation induced by CM CT26 in cachexia activation in myotubes. Firstly, we planned to block glycolysis to decrease the amount of pyruvate that is converted into lactate, by LDH. Glycolysis inhibition was obtained by using 2-deoxy-D-glucose (2-DG), that is a modified glucose molecule containing 2-hydroxyl group replaced by hydrogen that cannot undergoes further enzymatic modifications. Hence, myotubes were treated with CM CT26 and CM 4T1 (with or without 2-DG) for 24 h. The results show that glycolysis inhibition is effective in preventing the cachectic phenotype. Indeed, CM-CT26-treated myotubes containing 2-DG appear as control myofibers, as shown by images (Figure 2A) and myotube width (Figure 2B).

Coherently, immunoblot analysis demonstrates that glycolysis inhibition considerably reduces the high level of ubiquitinated proteins observed in CM-CT26-treated myotubes that becomes like that of the control and CM-4T1-treated myotubes (Figure 2C). Furthermore, glycolysis inhibition due to 2-DG prevents the decreased oxygen consumption (Figure 2D) and the increased lactate production (Figure 2E) in CM-CT26-treated myotubes.

To analyze the involvement of lactate production in CM-CT26-treated myotubes in cachexia activation, we impeded fermentation by using oxamate, the inhibitor of LDH. Myotubes were treated with CM CT26 or CM 4T1 (with or without oxamate) for 24 h. We used 75 mM oxamate that promotes the decrease of lactate production (Appendix A) without affecting cell viability (Appendix A). The results confirm the previous observations obtained by inhibiting glycolysis by using 2-DG.

Indeed, LDH inhibition prevents the CM-CT26-induced myotube cachectic phenotype and the reduction of myotube width (Figure 3A,B). In agreement, protein ubiquitination level due to CM CT26 treatment is greatly prevented by oxamate addition (Figure 3C). More interestingly, the absence in myotubes of the phenotypic and molecular cachectic effects induced by CM CT26 are associated with the lack of the metabolic alterations. Indeed, inhibition of lactate production by oxamate (Figure 3E) is associated with normal oxygen consumption in CM-CT26-treated myotubes in the presence of LDH inhibitor (Figure 3D).

These findings suggest that the metabolic shift towards fermentation in CM-CT26-treated myotubes could be crucially involved in the activation of cachexia, since the abolishment of CM-CT26-mediated metabolic switch, through glycolysis block or LDH inhibition, prevents the onset of the cachectic phenotype.

### 3.3. Inhibition of Glycolysis or Lactate Production Restores PDH Activity and Mitochondrial Membrane Potential in Cachectic Myotubes

We wondered whether 2-DG and oxamate can abolish the alteration of the mitochondrial membrane potential observed in CM-CT26-treated myotubes.

The analysis of mitochondrial potential was carried out by confocal analysis, using TMRM as a probe. The results show that the block of the metabolic shift associated with the induction of cachexia in CM-CT26-treated myotubes restores the normal membrane potential in mitochondria (Figure 4A,B).

Moreover, we analyzed whether the treatment of myotubes with CM CT26 alone or supplemented with 2-DG or oxamate affects the activity of the pyruvate dehydrogenase (PDH). PDH is one of the three enzymatic subunits (E1, E2, and E3) forming the pyruvate dehydrogenase complex (PDC) that catalyzes the conversion of pyruvate in Acetyl-CoA. Particularly, PDH (corresponding to E1 subunit) catalyzed pyruvate decarboxylation which is the first step in the conversion of pyruvate in Acetyl-CoA [22]. It has been reported that PDH activity is greatly decreased in cachectic conditions [23]. In agreement with published results, we observed that PDH activity is decreased in CM-CT26-treated myotubes, while the treatment with 2-DG or oxamate restores PDH activity similar to the control and CM-4T1-treated myotubes (Figure 3C). In addition, no treatment modifies the level of expression of the E1 subunit, which remains like control myotubes (Figure 3D).

Our findings show that inhibition of glycolysis by 2-DG or lactate production by oxamate has consequences at the mitochondrial level, where PDH activity and mitochondrial membrane potential are reported at the level of control myotubes.

### 3.4. IL-6 Is Involved in the Metabolic Change Leading to Cachectic Myotubes

Cachectic condition is characterized by a high production of pro-inflamed cytokines as IL-6 [12]. Immunoblot analysis shows that CM CT26 displays high amount of IL-6 in comparison to CM 4T1 (Figure 5A). This observation could suggest an involvement of IL-6 in the metabolic modification occurring in CM-CT26-treated myotubes leading to cachectic phenotype. To verify this hypothesis, we treated myotubes with IL-6 (120 ng/mL), and then we analyzed myotube phenotype and metabolism. The addition of IL-6 induces a cachectic phenotype characterized by thinner myotubes, as shown by microscopic images (Figure 5B), and myotube width reduction (Figure 5C). Furthermore, IL-6 displays the same metabolic changes due to CM CT26, namely decreased OCR (Figure 5D) and increased lactate production (Figure 5E). More interestingly, the cachectic phenotype is associated with metabolic change. Indeed, the concomitant treatment of myotubes with IL-6 and LDH inhibitor, oxamate, prevents cachectic phenotype (Figure 5B,C) and inhibits metabolic changes. Indeed, myotubes treated with IL-6 and oxamate show OCR and lactate production as control myotubes (Figure 5D,E). In addition to IL-6, previous results reported the involvement of Interferon-γ (IFN-γ) in cancer cachexia [24]. After proving that CM CT26 contains IFN-γ (Appendix A), myotubes are treated with IFN-γ (120 ng/mL) for 24 h. Our results show that IFN-γ induces a cachectic phenotype in myotubes as demonstrated by the analysis of the phenotypic effect (Appendix A), the measure of myotube width (Appendix A) and the level of protein ubiquitination (Appendix A). Furthermore, OCR analysis (Appendix A) and lactate amount (Appendix A) demonstrate that IFN-γ induces the metabolic change already observed by treating myotubes with CM CT26 and IL-6.

These findings suggest that IL-6 and IFN-γ are involved in the metabolic changes that leads to cachexia activation in myotubes.

### 3.5. CM CT26 Removal Restores Phenotype and Metabolism of Control Myotubes

We wondered if the phenotypic and metabolic modifications induced in myotubes by CM CT26 could be reverted. Hence, after the treatment of myotubes for 24 h with CMs, CMs were removed and myotubes were replenished with differentiating medium for additional 48 h. We observed that CM CT26 removal reverts cachectic phenotype as shown by images (Figure 6A), myotube width (Figure 6B) and decreased protein ubiquitination (Figure 6C). Interestingly, the reversion of cachectic phenotype is associated with the return of control metabolism. Indeed, OCR and lactate amount, that appeared decreased and increased, respectively, in CM-CT26-treated myotubes return to control values after CM CT26 removal (Figure 6D,E).

These findings suggest that induction of cachexia in myotubes could be reverted, at least in the early phase of the process.

### 3.6. CMs from Colon and Pancreatic Human Carcinoma Cell Lines Induce Lactate Production by Myotubes Associated to Cachectic Phenotype

We planned to verify whether the involvement of the metabolic change observed in myotubes due to CM CT26 treatment and leading to cachectic phenotype, could be observed by using CMs from human cancer cell lines. For this aim, we chose human carcinoma cell lines obtained from human tumors that are not able or able to trigger cachexia in vivo. We used colon carcinoma HCT116 and pancreatic carcinoma MiaPaCa-2 as human carcinoma frequently associated to cachexia, and as a counterpart, we used ovarian A2780 cells, breast carcinoma MDAMB231 cells and melanoma A375 cells. CMs obtained by HCT116 and MIAPaCa-2 cells induce cachectic phenotype in myotubes (Appendix A), as demonstrated by the decrease of myotube width (Appendix A) and the increase of the protein ubiquitination level (Figure 7A, left). As already observed with CM CT26, CM HCT116, and CM MIAPaCa-2 induces metabolic changes in treated myotubes. Myotubes treated with both CMs show decreased OCR (Figure 7B, left) and increased lactate production (Figure 7C, left). Interestingly, the addition of the LDH inhibitor oxamate to CM HCT116 and CM MIAPaCa-2 clearly reverts cachectic phenotype and metabolic effects induced by both CMs on myotubes, as already observed for CM CT26. In the presence of oxamate, CM-HCT116-treated or CM-MIAPaCa-2-treated myotubes are similar, for both the phenotype and the metabolism, to control and to myotubes treated with CM from non-inducing cachexia carcinoma cells (namely CM A2780, CM MDAMB231, and CM A375). Lactate production inhibition due to oxamate (Figure 7C, right) blocks decreased of myotube width (Appendix A, right), the increase of ubiquitination (Figure 7A, right) and the decrease of oxygen consumption (Figure 7B, right).

These findings demonstrate that CM from human carcinoma cells obtained by tumors able to induce cachexia in vivo also provokes the same cachectic effect in myotubes observed with murine CM CT26 and that cachexia activation is strictly dependent by a metabolic change.

## 4. Discussion

Cachexia is a disorder characterized by the uncontrolled loss of body weight associated with loss of homeostatic control of both energy and protein balance [2]. Worldwide, about 8.2 million people die from cancer each year, and at least half of the deaths are attributed to the tumors most frequently associated with cachexia, such as pancreatic, esophageal, gastric, pulmonary, hepatic, and colorectal cancers [25].

The present paper evidence that cachexia is induced in myotubes by a metabolic shift from oxidative to fermentative metabolism. Our findings demonstrate that cachectic myotubes secrete a higher lactate amount in comparison to control myotubes and the enhanced lactate production is sustained by an increased glucose uptake. The behavior of cachectic myotubes is very similar to that of cancer cells having a distinctive metabolism (known as the Warburg effect) characterized by the production of lactate in aerobic condition [26]. Our results reveal that CMs from carcinoma cell lines frequently associated with the onset of cachexia induce, in myotubes, a sort of Warburg effect, since treated myotubes have enhanced glucose uptake and activation of lactic fermentation in aerobic condition, as cancer cells do. This suggests that CMs obtained from cancer cells metabolically transform healthy myotubes into “cancer myotubes”. Myotubes treated with CMs from human colon and pancreatic carcinoma cell lines show the same metabolic change, thus suggesting that this metabolic shift is a general phenomenon. These data are in agreement with those reported by Der-Torossian et al., showing that gastrocnemius of cachectic mice displays an increased glycolytic pathway [27].

In our model of cachexia, cachectic myotubes display altered mitochondria, showing altered membrane potential and impaired oxygen consumption. These findings agree with previous results showing impaired mitochondrial structure in the skeletal muscle of animal models of cancer cachexia [28,29,30]. Furthermore, cancer patients have mitochondria in skeletal muscle with significant abnormalities that lead to compromised mitochondrial functions, as decreased oxidative capacity [31,32], modification of mitochondrial membrane [31], and oxidation of mitochondrial proteins [33].

Cancer cachexia is associated with several metabolic abnormalities [34]. A metabolomics approach has been used to analyze the metabolic perturbations in murine models of cancer-induced and chemotherapy-induced cachexia. The results showed that both models of cachexia are characterized by a decreased level of circulating glucose, due to an increased systemic glucose demand, depletion of hepatic glucose, and alterations in the flux through the tricarboxylic acid cycle and β-oxidation pathways [35]. Furthermore, inflammatory factors secreted by cancer-cell-inducing cachexia promote β-oxidation, leading to oxidative stress which impairs muscle growth [36]. However, no findings have demonstrated a direct and causal involvement of metabolic changes in cancer cachexia activation so far.

Inflammatory mediators, such as IL-6, play a crucial role in muscle wasting [24] and in cancer patients high circulating IL-6 levels correlate with weight loss and survival [20]. In addition, IL-6 signaling is involved in decreased myotube width during muscle wasting, and the IL-6-activated-STAT3 pathway plays a crucial role in the process [13,36]. Moreover, elevated amounts of pro-inflammatory cytokines, including IL-6, altered mitochondrial homeostasis, leading to dysfunction characterized by abnormal levels of mitochondrial reactive oxygen species and decreased ATP production [37]. Our findings show that CM CT26 contains a higher amount of IL-6 in comparison to CM 4T1, and the addition of IL-6 to healthy myotubes induces a cachectic phenotype in association with the increased lactate formation and decreased oxygen consumption. Moreover, IFN-γ, whose involvement in cancer cachexia has been previously reported [23], can induce the metabolic shift. These data demonstrate that both IL-6 and IFN-γ participate in the metabolic shift that leads to cancer cachexia activation.

It is intriguing that cachexia is not induced when glycolysis or LDH activity is blocked. Indeed, the inhibition of lactate production (either blocking glycolysis or LDH activity) impedes the formation of the cachectic features. At the mitochondrial level, the inhibition of lactate production restores mitochondrial membrane potential, PDH activity, and oxygen consumption similar to control myotubes. The comprehension of the mechanism involved in blocking cachectic myotube formation due to glycolysis or LDH inhibition is not easy to suggest. It has been reported that IL-6 is involved in muscle metabolism. For example, IL-6 enhances glucose uptake in muscle [38] and probably could be involved in the enhancement of glucose assumption observed in CM-CT26-treated myotubes. In addition, IL-6 downregulates the activity of the pyruvate dehydrogenase (PDH) complex in muscle [39]. PDH catalyzes the decarboxylation of the pyruvate, which is the first reaction in PDC that promotes the irreversible conversion of pyruvate to Acetyl-CoA, which has been used in the first reaction of Krebs cycle [40,41]. We found that the treatment of myotubes with CM CT26 decreases PDH activity, thus blocking mitochondrial metabolism, and forces the use of pyruvate obtained by glycolysis for lactic fermentation. The inhibition of glycolysis by 2-DG or LDH activity by oxamate restores PDH activity as control myotubes. This is likely due to a greater availability of pyruvate in myotubes treated with the two inhibitors. The question to be addressed is the source of pyruvate used by PDH. For what oxamate is concerned, pyruvate formed by glycolysis could be completely driven towards PDH. More complicated is the 2-DG mechanism. The analogue of glucose 2-DG is converted in 2-DG-6 phosphate that inhibits both phospho-glucose isomerase, in a competitive manner, and hexokinase, through a non-competitive mechanism, with a reduced formation of pyruvate in both cases [42]. Hence, in 2-DG-treated myotubes, pyruvate could originate from glucogenic amino acids, glycerol obtained from triglyceride degradation, and glyceraldeide-3 phosphate produced in the pentose phosphate pathway (that appears upregulated in 2-DG-treated cells) [42]. Whatever the source of pyruvate, in 2-DG- and oxamate-treated myotubes, PDH could be re-established, thus permitting the conversion of pyruvate in Acetyl-CoA and the restoration of oxygen consumption similar to control myotubes.

However, a role of the pyruvate dehydrogenase kinase 4 (PDK4) in the PDH reactivation cannot be excluded. Recently, a direct role for PDK4 in promoting cancer-associated muscle metabolic alterations and skeletal muscle atrophy has been reported, and a future study about this enzyme is mandatory [23]. The reactivation of PDH is associated with the acquisition of a normal mitochondrial membrane potential, sign of a re-acquired mitochondrial function, also testified by the oxygen consumption that returns to control levels.

It is important to underline that lactate production in cachectic myotubes occurs in aerobic conditions, while healthy skeletal muscle activates lactic fermentation in anaerobic conditions, thus establishing with the liver the so-called Cori cycle. After inflammatory cytokine stimulation (i.e., IL-6), healthy myotubes shift towards fermentation due to altered mitochondria. The enhanced glucose uptake could allow myotubes to produce ATP from glycolysis necessary for survival. From this point of view, the metabolic change towards fermentation could be a mechanism to counteract cell death. In agreement with this hypothesis, we observe that myotubes after 24 h of treatment with CM CT26 are viable and can revert cachectic phenotype following CM CT26 removal.

Without any doubts, cachexia negatively affects daily life and care of cancer patients. Several efforts have been carried out to counteract the onset of pathology. These include a combination of specific nutritional supports associated with pharmacological interventions and muscle exercise [43]. Drugs showing beneficial effects are megestrol acetate, ghrelin agonists, and melanocortin 4 (MC4) receptor antagonists [44]. Next to these nutritional approaches, impeding the formation of lactate by inhibiting LDH could also be important in cancer cachexia. The final product of LDH, lactate, has been considered for a long time only a “waste product” of aerobic glycolysis. To date, the involvement of lactate in tumor progression, as cell migration and metastasis formation and the use of lactate as energy source has demonstrated [45]. Hence, LDH inhibition constitutes a powerful tool to counteract cancer progression and those pathologies correlated to cancer, as cachexia. LDH level (serum and/or intracellular) has been found altered in several types of cancers (as hematopoietic malignancies, lung, and breast cancers), and the use of LDH as a possible marker and therapeutic target has been suggested [46]. Although several in vivo studies will be necessary to define the role of LDH in cancer cachexia, our findings open a new possibility of treatment of this pathology.

## Figures and Tables

**Figure 1 biomedicines-09-00698-f001:**
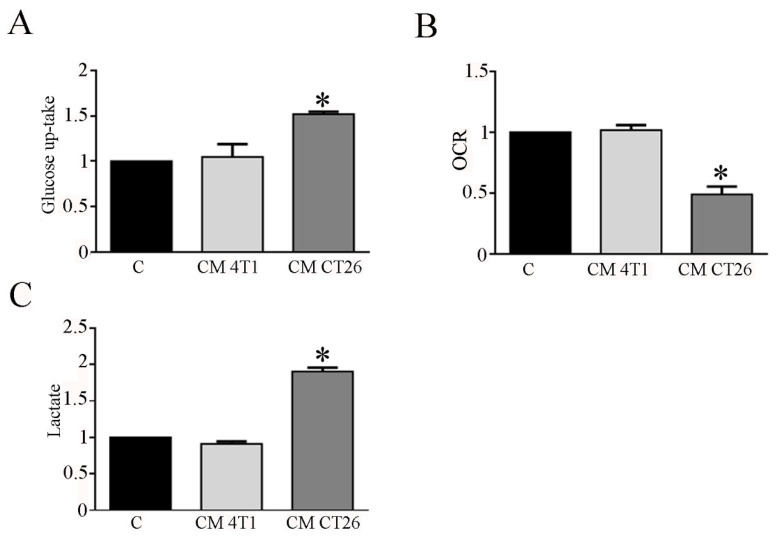
CM CT26 treatment modifies glucose metabolism in myotubes. Twenty-four hours after the treatment of myotubes with CMs or with differentiating medium (reported as C, control), (**A**) glucose uptake, (**B**) Oxygen Consumption Rate (OCR), and (**C**) extracellular lactate amount was assayed. The values are reported as folding increase considering control myotubes as 1; n = 4; * *p* < 0.05.

**Figure 2 biomedicines-09-00698-f002:**
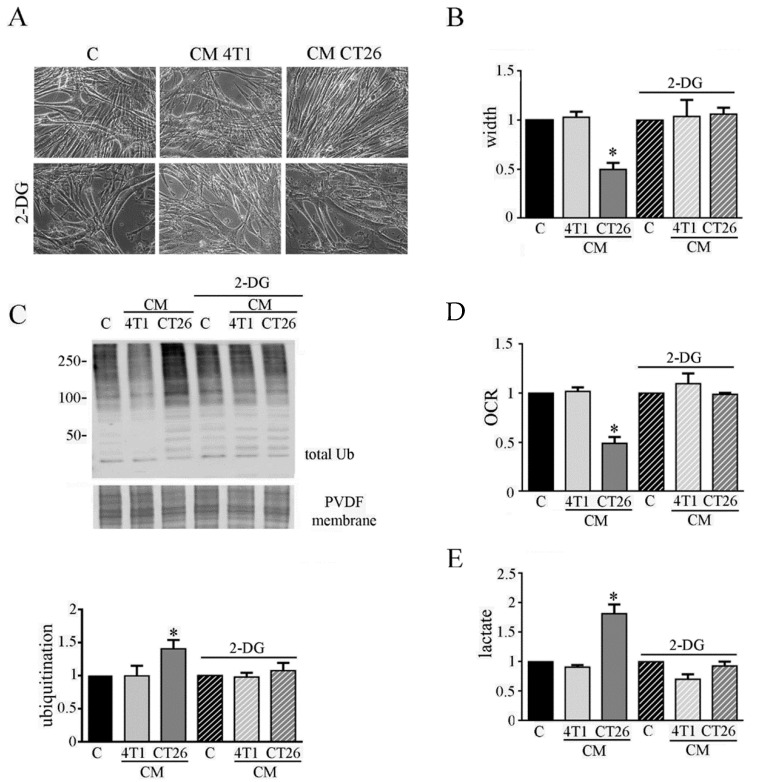
Inhibition of glycolysis impairs cachexia in myotubes. Four days-differentiated myotubes were treated with CM 4T1 or CM CT26 or differentiating medium (C, control) for 24 h. Where indicated, 2-deoxy-glucose (2-DG) (1 mg/mL final) was added to media. (**A**) Representative optical microscope images of treated myotubes with or without 2-DG. Scale bar: 100 µm. (**B**) Measure of myotube width 24 h after the treatment. (**C**) Ubiquitination level of myotubes. Total ubiquitination level reported in the bar graph was obtained by using Coomassie-stained PVDF membrane for normalization. (**D**) Analysis of oxygen consumption rate (OCR). (**E**) Assay of lactate amount in myotubes. All the values in the bar graphs are reported as fold increase, considering control myotubes as 1; n = 4; * *p* < 0.05.

**Figure 3 biomedicines-09-00698-f003:**
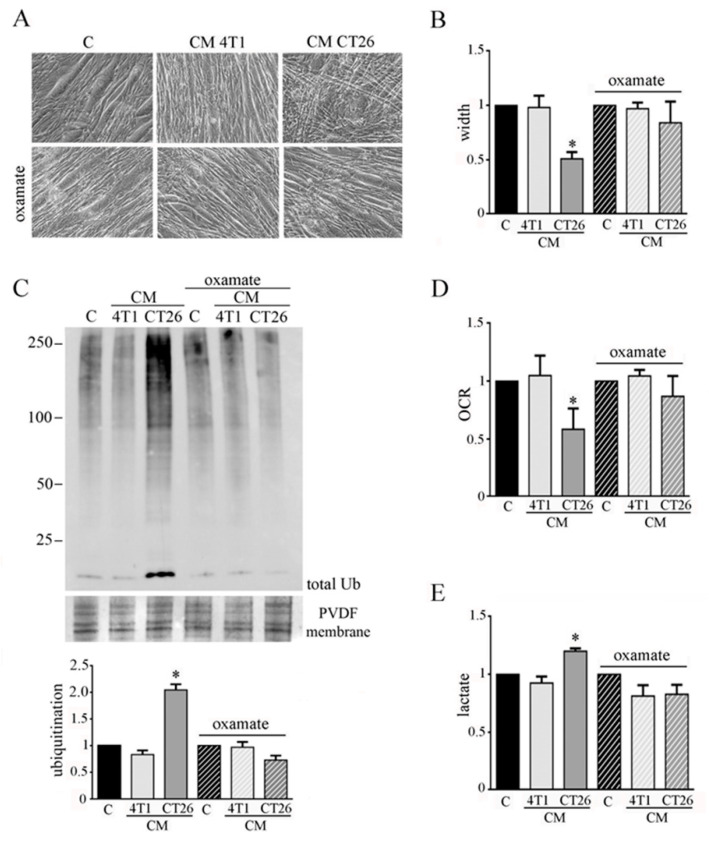
LDH inhibition impedes CM-CT26-induced cachexia. Four-day-differentiated myotubes were treated with CMs figure 24 h. Untreated myotubes (indicated with C) are maintained in differentiating medium for the same period. Where indicated, the LDH inhibitor’s oxamate (75 mM final) was added to the culture media. (**A**) Representative optical microscope images of CM-treated myotubes in the presence of oxamate. Scale bar: 100 µm. (**B**) Analysis of oxamate effect in myotube width. (**C**) Ubiquitination level in myotubes. Total ubiquitination level, reported in the bar graph, was obtained by using Coomassie-stained PVDF membrane for normalization. (**D**) Effect of oxamate in oxygen consumption rate (OCR) of CM-CT26-treated myotubes. (**E**) Lactate production assay in CM-treated myotubes in the presence of oxamate. All the values in the bar graphs are reported as fold increase, considering control myotubes as 1; n = 5; * *p* < 0.05.

**Figure 4 biomedicines-09-00698-f004:**
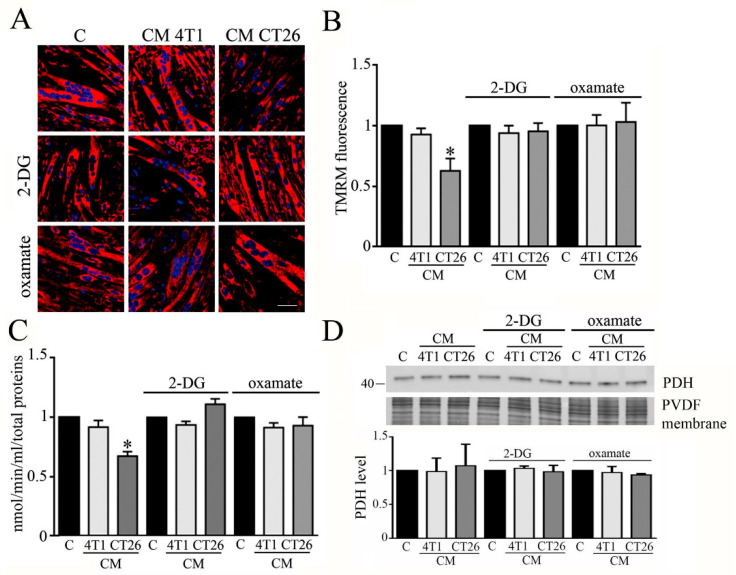
Analysis of mitochondrial membrane potential and PDH in myotubes treated with CMs contained 2-DG or oxamate. Four-day-differentiated myotubes were treated with CMs for 24 h or maintained in differentiating medium as C (control). Where indicated, 2-deoxy-glucose (2-DG) (1 mg/mL final) or oxamate (75 mM final) was added to the media. (**A**) Mitochondria have been labeled with TMRM (1 µm final) and immediately analyzed under confocal microscope. Representative images of myotubes labeled with TMRM. Mitochondria appear red, while nuclei are blue due to DAPI staining. (**B**) TMRM fluorescence reported as the mean fluorescence in at least ten randomly chosen fields. Scale bar: 50 µm. (**C**) PDH activity. PDH activity measured as nmol/min/mL is normalized by using total protein content in each sample. (**D**) PDH immunoblot. PDH level, reported in the bar graph, was obtained by using Coomassie-stained PVDF membrane for normalization. All the values in the bar graphs are reported as fold increase, considering control myotubes as 1; n = 4; * *p* < 0.05.

**Figure 5 biomedicines-09-00698-f005:**
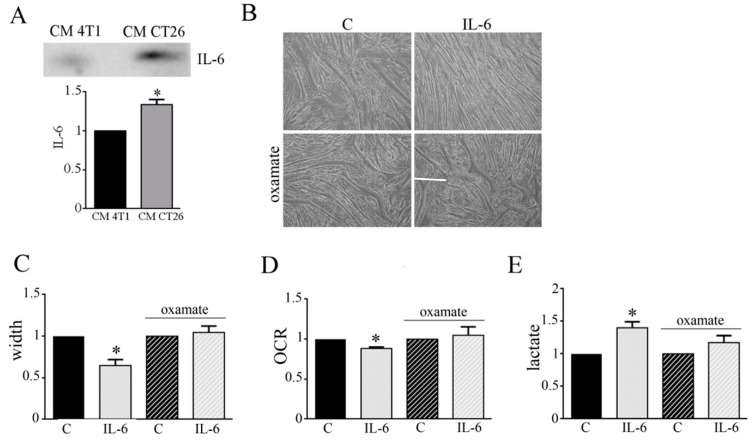
IL-6 secreted by CT26 colon carcinoma cells participates in metabolic shift leading to cachexia. (**A**) Anti-IL-6 immunoblot analysis performed in CM 4T1 and CM CT26. IL-6 level was normalized on CM protein content. Four-day-differentiated myotubes were treated with IL-6 (120 ng/mL) for 24 h, except untreated myotubes (indicated with C) that are maintained in differentiating medium for the same period. Where indicated, the LDH inhibitor’s oxamate (75 mM final) was added to the culture media. (**B**) Phenotypic effect due to IL-6 treatment in myotubes observed by optical microscope. Scale bar: 100 µm. (**C**) Measure of myotube width in the presence of IL-6. (**D**) Oxygen consumption rate (OCR) analysis and (**E**) lactate assay. All the values in the bar graphs are reported as fold increase, considering control myotubes as 1; n = 4; * *p* < 0.05.

**Figure 6 biomedicines-09-00698-f006:**
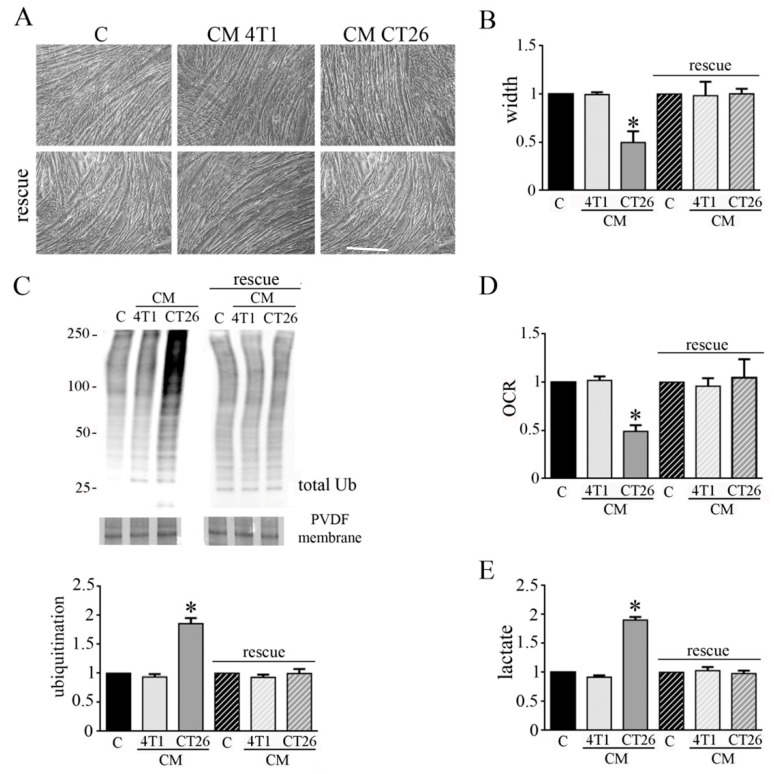
CM CT26 removal restores normal phenotype and metabolism in myotubes. Four-day-differentiated myotubes were treated with 24 h with CM 4T1 or CM CT26. CMs were then removed, and differentiating medium was added to myotubes for additional 48 h. Control myotubes, indicated as C, were treated with differentiating medium throughout the experiment. (**A**) Microscopic images of myotubes before CM removal (images above) and after the treatment with differentiating medium (images below). Scale bar: 100 µm. (**B**) Measure of myotube width. (**C**) Immunoblot of ubiquitinated proteins in myotubes. Normalized ubiquitination level was obtained by using Coomassie-stained PVDF membrane. (**D**) Oxygen consumption rate (OCR) analysis. (**E**) Lactate assay. All the values in the bar graphs are reported as fold increase, considering control myotubes as 1; n = 4; * *p* < 0.05.

**Figure 7 biomedicines-09-00698-f007:**
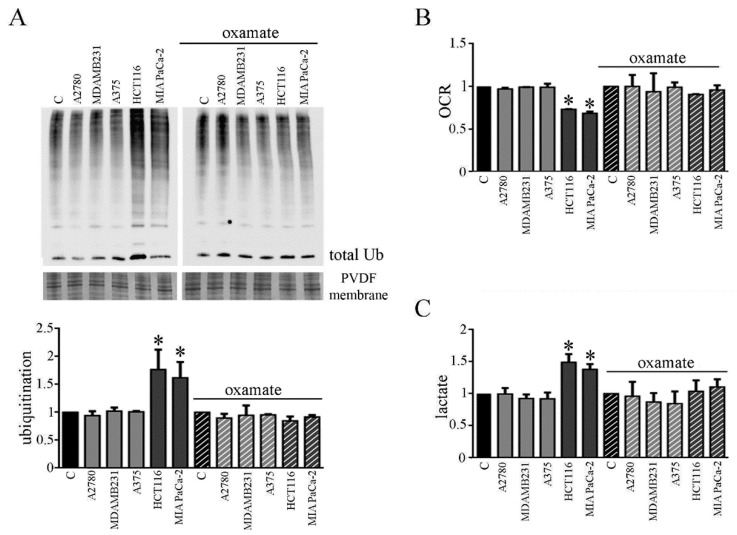
LDH inhibition in myotubes treated with CM from colon and pancreatic human carcinoma cell lines prevents induction of cachexia. CMs were obtained by cultivating human carcinoma cells not-inducing cachexia in vivo (breast carcinoma MDAMB231, melanoma A375, ovarian A2780) and human carcinoma cells inducing cachexia in vivo (colon carcinoma HCT116, pancreatic carcinoma MIAPaCa-2). Four-day-differentiated myotubes were treated with CMs for 24 h. Untreated myotubes (indicated with C) were maintained in differentiating medium for the same period. Where indicated, LDH inhibitor’s oxamate (75 mM final) was added to media. (**A**) Ubiquitination level in myotubes treated with different media with or without oxamate. Bar graph shows total ubiquitination level obtained by using Coomassie-stained PVDF membrane for normalization. (**B**) Oxygen consumption rate (OCR) analysis in treated myotubes with or without oxamate. (**C**) Assay of lactate in the media of CM-treated myotubes with or without oxamate. All the values in the bar graphs were reported as fold increase, considering control myotubes as 1; n = 3; * *p* < 0.05.

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
