# Peer review of "A Metabolic Change towards Fermentation Drives Cancer Cachexia in Myotubes"

_biomedicines, 2021, doi:10.3390/biomedicines9060698_

Round 1

Reviewer 1 Report

The inclusion of the PDH activity assay added an important mechanistic feature to the paper.  It is interesting that both 2DG and oxamate lead to a restoration of PDH activity.  It must be recognized that the mechanisms by which these inhibitors impact metabolism are likely to be quite different.  In the discussion, the authors suggest that these inhibitors would make pyruvate more available for PDH.  This is plausible when oxamate is used as pyruvate can no longer make be reduced to lactate by LDH.  It is not clear how the use of 2DG could make pyruvate more available.  The classical explanation of 2DG activity is that it blocks the glycolytic reactions beyond 2-DG-6P.  This would lead to less downstream availability of pyruvate.  The mechanisms by which 2DG affect metabolism are indeed more complicated than that.  The paper by Ralser et al (PNAS, 105 (45) 17807, 2008) does a nice job explaining other potential mechanisms such as effects on glycosylation pathways, alterations to the pentose phosphate pathway, etc.  Would be useful for the authors to expand their discussion to acknowledge that these inhibitors of glycolysis have different mechanism.  Their consistent impact on PDH is interesting and determining the mechanisms by which each of them impacts cachexia will require further study.

Author Response

Point to point answer to referee 1

The inclusion of the PDH activity assay added an important mechanistic feature to the paper.  It is interesting that both 2DG and oxamate lead to a restoration of PDH activity.  It must be recognized that the mechanisms by which these inhibitors impact metabolism are likely to be quite different.  In the discussion, the authors suggest that these inhibitors would make pyruvate more available for PDH.  This is plausible when oxamate is used as pyruvate can no longer make be reduced to lactate by LDH.  It is not clear how the use of 2DG could make pyruvate more available.  The classical explanation of 2DG activity is that it blocks the glycolytic reactions beyond 2-DG-6P.  This would lead to less downstream availability of pyruvate.  The mechanisms by which 2DG affect metabolism are indeed more complicated than that.  The paper by Ralser et al (PNAS, 105 (45) 17807, 2008) does a nice job explaining other potential mechanisms such as effects on glycosylation pathways, alterations to the pentose phosphate pathway, etc.  Would be useful for the authors to expand their discussion to acknowledge that these inhibitors of glycolysis have different mechanism.  Their consistent impact on PDH is interesting and determining the mechanisms by which each of them impacts cachexia will require further study.

We thank the referee for the suggestion. As requested, we increased the discussion about PDH regarding the different mechanisms between 2-DG and oxamate, as follows (lines 419-430):

“This is likely due to a greater availability of pyruvate in myotubes treated with the two inhibitors. The question to be addressed is the source of pyruvate used by PDH. For what oxamate is concerned, pyruvate formed by glycolysis could be completely driven towards PDH. More complicated is the 2-DG mechanism. The analogue of glucose 2-DG is converted in 2-DG-6 phosphate that inhibits both phospho-glucose isomerase, in a competitive manner, and hexokinase, through a non-competitive mechanism, with a reduced formation of pyruvate in both cases [42]. Hence, in 2-DG-treated myotubes, pyruvate could originate from glucogenic amino acids, glycerol obtained from triglyceride degradation and glyceraldeide-3 phosphate produced in the pentose phosphate pathway (that appears up-regulated in 2-DG-treated cells) [42]. Whatever the source of pyruvate, in 2-DG and oxamate treated myotubes PDH could be re-established, thus permitting the conversion of pyruvate in Acetyl-CoA and the restoration of oxygen consumption like control myotubes.”

Moreover, we inserted the new reference #42 regarding the paper suggested by the referee.

  1. Ralsera M, Wamelinkb MM, Struys EA, et al. A catabolic block does not sufficiently explain how 2-deoxy-D-glucose inhibits cell growth. PNAS 2008; 46: 17807–17811.

Reviewer 2 Report

The additional data included in the present form of the manuscript further support the general conclusion of the paper. I have not other issues in addition to what I already highlighted in the previous submission.

Author Response

We thank the referee.

Below the response to referee 1.

Point to point answer to referee 1

The inclusion of the PDH activity assay added an important mechanistic feature to the paper.  It is interesting that both 2DG and oxamate lead to a restoration of PDH activity.  It must be recognized that the mechanisms by which these inhibitors impact metabolism are likely to be quite different.  In the discussion, the authors suggest that these inhibitors would make pyruvate more available for PDH.  This is plausible when oxamate is used as pyruvate can no longer make be reduced to lactate by LDH.  It is not clear how the use of 2DG could make pyruvate more available.  The classical explanation of 2DG activity is that it blocks the glycolytic reactions beyond 2-DG-6P.  This would lead to less downstream availability of pyruvate.  The mechanisms by which 2DG affect metabolism are indeed more complicated than that.  The paper by Ralser et al (PNAS, 105 (45) 17807, 2008) does a nice job explaining other potential mechanisms such as effects on glycosylation pathways, alterations to the pentose phosphate pathway, etc.  Would be useful for the authors to expand their discussion to acknowledge that these inhibitors of glycolysis have different mechanism.  Their consistent impact on PDH is interesting and determining the mechanisms by which each of them impacts cachexia will require further study.

We thank the referee for the suggestion. As requested, we increased the discussion about PDH regarding the different mechanisms between 2-DG and oxamate, as follows (lines 419-430):

“This is likely due to a greater availability of pyruvate in myotubes treated with the two inhibitors. The question to be addressed is the source of pyruvate used by PDH. For what oxamate is concerned, pyruvate formed by glycolysis could be completely driven towards PDH. More complicated is the 2-DG mechanism. The analogue of glucose 2-DG is converted in 2-DG-6 phosphate that inhibits both phospho-glucose isomerase, in a competitive manner, and hexokinase, through a non-competitive mechanism, with a reduced formation of pyruvate in both cases [42]. Hence, in 2-DG-treated myotubes, pyruvate could originate from glucogenic amino acids, glycerol obtained from triglyceride degradation and glyceraldeide-3 phosphate produced in the pentose phosphate pathway (that appears up-regulated in 2-DG-treated cells) [42]. Whatever the source of pyruvate, in 2-DG and oxamate treated myotubes PDH could be re-established, thus permitting the conversion of pyruvate in Acetyl-CoA and the restoration of oxygen consumption like control myotubes.”

Moreover, we inserted the new reference #42 regarding the paper suggested by the referee.

  1. Ralsera M, Wamelinkb MM, Struys EA, et al. A catabolic block does not sufficiently explain how 2-deoxy-D-glucose inhibits cell growth. PNAS 2008; 46: 17807–17811.

Reviewer 3 Report

The alterations and additional experiments performed by the authors have significantly improved the overall quality of the manuscript. I have no further comments.

Author Response

(The authors gave the same response as above.)
